# Structural insights into the conformational changes of BTR1/SLC4A11 in complex with PIP$_2$

Yishuo Lu [1,2,6], Peng Zuo [3,6], Hongyi Chen[1,2], Hui Shan[1], Weize Wang[2,3], Zonglin Dai[3], He Xu[4], Yayu Chen[4], Ling Liang[3,5], Dian Ding [1,3], Yan Jin[3] & Yuxin Yin [1,2,3] ✉

BTR1 (SLC4A11) is a NH$_3$ stimulated H$^+$ (OH$^-$) transporter belonging to the SLC4 family. Dysfunction of BTR1 leads to diseases such as congenital hereditary endothelial dystrophy (CHED) and Fuchs endothelial corneal dystrophy (FECD). However, the mechanistic basis of BTR1 activation by alkaline pH, transport activity regulation and pathogenic mutations remains elusive. Here, we present cryo-EM structures of human BTR1 in the outward-facing state in complex with its activating ligands PIP$_2$ and the inward-facing state with the pathogenic R125H mutation. We reveal that PIP$_2$ binds at the interface between the transmembrane domain and the N-terminal cytosolic domain of BTR1. Disruption of either the PIP$_2$ binding site or protonation of PIP$_2$ phosphate groups by acidic pH can transform BTR1 into an inward-facing conformation. Our results provide insights into the mechanisms of how the transport activity and conformation changes of BTR1 are regulated by PIP$_2$ binding and interaction of TMD and NTD.

SLC4 transporters play essential roles in regulating intracellular pH (pHi) and other physiological processes such as CO$_2$ transport by erythrocytes and solute secretion/reabsorption across epithelia[1]. SLC4A11 (or BTR1, NaBC1) is a member of the SLC4 transporter family which was first classified and named as bicarbonate transporter-related protein-1 (BTR1) because of its sequence homology with other SLC4 family transporters[2]. However, unlike other SLC4 family members, BTR1 does not transport bicarbonate[2,3]. In contrast, based on its sequence similarity with the Arabidopsis borate transporter BOR1, BTR1 was first thought to function as an electrogenic, voltage-regulated, Na$^+$-B(OH)$_4^-$ coupled transporter and was renamed as NaBC1[3]. BTR1 was subsequently shown to have NH$_3$/H$^+$ electrogenic cotransport activity, which was atypical among SLC4 bicarbonate transporters[4]. This NH$_3$/H(n)$^+$ co-transport activity of BTR1 was independently corroborated by subsequent investigations[5,6]. However, as

H$^+$ conductance of BTR1 is stimulated by alkaline pH and NH$_3$ is difficult to measure, whether NH$_3$ co-transport with H$^+$ is still in debate.

Like other SLC4 family transporters, BTR1 is widely expressed in kidney, salivary glands, testis, thyroid glands, and trachea[2,7,8]. In addition, BTR1 has been detected throughout the endothelial cell layer of cornea and the audio-vestibular system[9]. Given the uncertain influence of microelement borate homeostasis on the human body, apparent changes in pH$_i$ and osmotic pressure are more likely to underlie SLC4A11-mutation associated diseases such as congenital hereditary endothelial dystrophy (CHED)[10–12], Fuchs' endothelial corneal dystrophy (FECD)[13] and Harboyan syndrome (or corneal dystrophy and perceptive deafness, CDPD)[14]. More than 60 disease-associated mutations in SLC4A11 have been reported[15]. These diseases occur during different stages of life, but collectively, they are all associated with impaired vision or hearing. In addition to corneal transplantation, proposed

[1]Institute of Precision Medicine, Peking University Shenzhen Hospital, Shenzhen 518036, China. [2]Peking-Tsinghua Center for Life Sciences, Peking University, Beijing 100871, China. [3]Institute of Systems Biomedicine, Department of Pathology, Beijing Key Laboratory of Tumor Systems Biology, School of Basic Medical Sciences, Peking University Health Science Center, Beijing 100191, China. [4]XtalPi, Beijing, China. [5]Department of Biochemistry and Biophysics, School of Basic Medical Sciences, Peking University Health Science Center, Beijing 100191, China. [6]These authors contributed equally: Yishuo Lu, Peng Zuo. ✉e-mail: yinyuxin@hsc.pku.edu.cn

treatment involves the restoration of the ER mis-localized mutant BTR1 to its normal membrane localization[16]. However, there are currently no treatments available that target mutant BTR1 with altered function. For this reason, solving the structure of BTR1 and understanding the mechanism of its transport activity are extremely important for gaining insights into therapeutic strategies to approach certain BTR1-related diseases.

In recent years, several structures of SLC4 family transporters have been reported, such as the crystal and cryo-EM structures of human[17–19], bovine[20] AE1 (SLC4A1), recently solved cryo-EM structures of AE2 (SLC4A2)[21], cryo-EM structures of NBCe1 (SLC4A4)[22] and NDCBE (SLC4A8)[23]. These structures describe the protein interactions[18,19], the substrates binding sites[17,18,21–23], the inhibitor binding sites[17,21] and provide explanations for the consequences of certain pathogenic mutations[17,18,22] and the regulatory mechanism of transport activity[21]. However, these structures did not resolve the complete NH2-terminal domains (NTDs), which leaves a critical gap in our understanding of the interaction(s) between the NTDs and TMDs of these SLC4 transporters. These prompted us to explore the relationship between the NTD and TMD of SLC4 transporters, and how it is influenced by pH.

Here, we have used single-particle cryo-EM to determine human BTR1 full-length structures in the outward-facing states with and without addition of ammonia at resolutions of 2.94 Å and 2.84 Å, respectively. We also describe the inward-facing structures of BTR1 containing the pathological mutation R125H, and BTR1 under acidic pH conditions at resolutions of 2.96 Å and 2.94 Å, respectively. These structures reveal that the interaction between the NTD and TMD of BTR1 in the outward-facing state is mainly mediated by PIP2, the key

regulator of BTR1 activity, as well as how disease-associated mutations disrupt BTR1 function. Our work provides insights into the regulatory mechanisms of BTR1 and other SLC4 transporters.

## Results

### Cryo-EM structure of BTR1 in the outward-facing state

For structural and electrophysiological studies, full-length human BTR1, fused with a C-terminal green fluorescent protein (GFP) and a His8 tag, was expressed in FreeStyle HEK293F cells[24]. Whole-cell patch clamp approach was used to measure the $H^+$ conductance of cells transfected with recombinant BTR1 or empty vector (EV) (Fig. 1a and Supplementary Fig. 1). No currents were observed in cells transfected with EV during $NH_4Cl$ exposures (Supplementary Fig. 1a–d) whereas $NH_4Cl$ exposures induced apparent inward currents of cells transfected with recombinant BTR1 (Fig. 1a and Supplementary Fig. 1e, f) accompanied by a shift in zero-current potentials from −17 mV to −4 mV (Fig. 1a), which is similar as previous study[6]. Recombinant BTR1 overexpressed in FreeStyle HEK293F cells was purified in Glyco-diosgenin (GDN) detergent-solubilized micelles (Supplementary Fig. 1g, h) and then subjected to cryo-EM analysis. Structures of BTR1 with the addition of $NH_4Cl$ (BTR1$_{OF/NH3}$ state) and in the apo state (BTR1$_{OF/APO}$ state) were resolved at resolutions of 2.84 Å and 2.94 Å, respectively (Supplementary Fig. 2).

These BTR1 structures reveal an outward-facing dimeric architecture with each monomer containing a TMD (residues 336–891) and a cytoplasmic domain (NTD) (residues 1–307). The TMD contains a core domain (comprising TMs 1–4 and 8–11) and a gate domain (comprising TMs 5–7 and 12–14) (Fig. 1b), resembling the arrangement

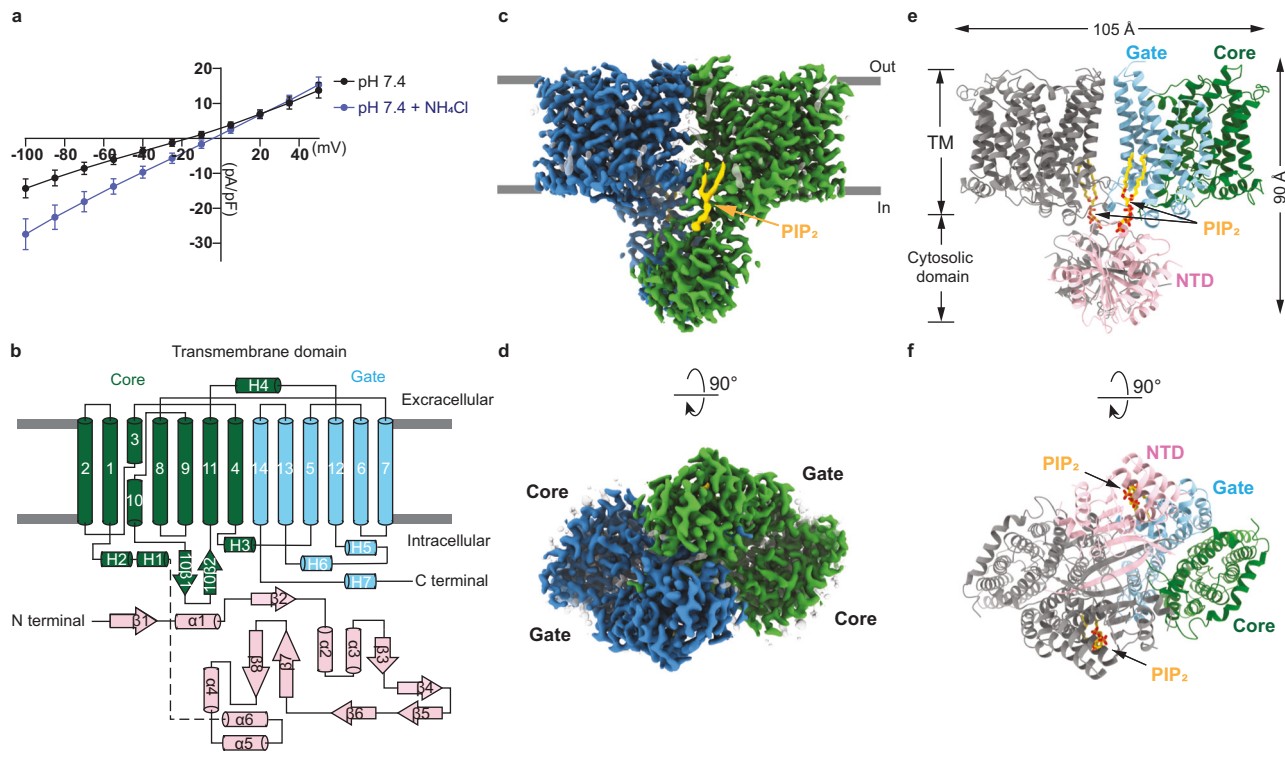

**Fig. 1 | Cryo-EM structure of BTR1 in the outward-facing state. a** I−V curve of cells expressing wild-type BTR1 at pH 7.4 in the presence and absence of 5 mM $NH_4Cl$. The current values are standardized by cell capacitance. Data shown are mean values ± s.d. of *n* = 4 biologically independent experiments. **b** Topology and domain arrangement of a BTR1 monomer. The gate domain, core domain and N-terminal cytoplasmic domain (NTD) are shown in blue, green and pink, respectively. Gray bars represent the boundaries of the cell membrane. **c, d** Cryo-EM maps

of the outward-facing state of a BTR1 dimer bound to PIP2: views from the side (**c**) and the bottom (**d**). The two monomers of BTR1 are colored in blue and green, and the PIP2 molecules are colored in gold. The remaining lipid molecules are shown as transparent gray. **e, f** Structural model of the outward-facing state BTR1 views from the side (**e**) and the bottom (**f**). The two PIP2 molecules are shown in stick representations. The color scheme of one BTR1 monomer is the same as in (**b**), and the other monomer is colored in gray.

of UraA[25]. In the outward-facing state, the BTR1 dimer occupies a 3D space of ~105 Å × 65 Å × 90 Å (Fig. 1c–f). The TMD of BTR1 in the outward-facing state closely resembles the structures of SLC4 family members AE1 (pdb 4YZF) (Supplementary Fig. 3a)[17], NBCe1 (pdb 6CAA) (Supplementary Fig. 3b)[22] and NDCBE (pdb 7RTM) (Supplementary Fig. 3c)[23] with RMSDs of 1.860 Å, 1.780 Å and 2.180 Å, respectively. The formation of a dimeric TMD mainly depends on the interaction of the loop between H5 and H6 helix of one monomer and the loop between TM6 and TM7 of the other monomer in the gate domain (Supplementary Fig. 3d). In addition, the hydrophobic interactions occur in TM5, TM6, and TM7 of the two monomers also promote the formation of dimeric TMDs (Supplementary Fig. 3e). The formation of dimeric NTDs mainly relies on the hydrogen bonds between the β1 sheets of the two monomers (Supplementary Fig. 3f). Finally, the TMD and NTD interact through the electrostatic interactions of $PIP_2$ with the residues in the $PIP_2$ binding pocket. Additionally, the hydrophobic interaction between I124 and P825, as well as the electrostatic interactions between E122 and R823, E123 and R822 assist in the association of the TMD and NTD (Supplementary Fig. 3g).

## Substrate binding sites of BTR1

Previous studies proposed that BTR1 is a $H^+$ transporter[4–6]. BTR1 shares a similar membrane topology with other SLC4 family members, with its substrate binding sites located in the central cavity between the gate and core domains. Negative charges in TM1, TM3, TM5, TM8 and TM10 suggest that residues in these helices contribute to the $H^+$ binding site (Fig. 2a, b). We analysed the pore dimensions of the transmembrane domains using HOLE2[26]. The results show that the substrate binding sites of $BTR1_{OF/APO}$ and $BTR1_{OF/NH3}$ are accessible from the extracellular side (Fig. 2c, d), confirming their outward-facing conformation. In the cryo-EM map of $BTR1_{OF/NH3}$ state, we identified two densities with relatively different intensity from the $BTR1_{OF/APO}$ state in the substrate binding sites and both states reveal several extra densities (represented by Molecule 1–6), distribute along TM5 and the ion permeation pathway lined up by residues F393, L396, G509, D505, T501 and I498 (Supplementary Fig. 3h–j). These densities could be either waters or ammonia. However, lack of details of the densities precludes the further determination of their identities. In our BTR1 outward-facing state model, pathogenic mutations H724A and E675Q[15,27,28] are located at the substrate binding site and near the extra densities (Supplementary Fig. 3h). E675 of BTR1 is partly conserved among human SLC4 family members and BTR1 orthologs (Fig. 2e–h, j). M3 (Molecule 3) could be fixed by the formation of hydrogen bonds between the side chain of E675 and the backbones of A720 and Y722 (Supplementary Fig. 3h). This negatively charged amino acid may participate in the coordination of $H^+$ during substrates transport process of BTR1, as the corresponding D800 of NDCBE forms ionic bond with $Na^+$ (Fig. 2h) and the substitution of this residue significantly impairs the activity of SLC4 family transporters[17,22,23]. P437 is highly conserved among human SLC4 transporters and BTR1 orthologs, but P723, H719 and H724 are only conserved among BTR1 orthologs (Fig. 2e–h, j), which may be responsible for the atypical substrate selectivity of BTR1. M5 and M6 are both located at the substrate binding pocket and M5 could form hydrogen bond with the backbone carbonyl group of T434 (Supplementary Fig. 3h). M2 at the entrance of the binding pocket may form hydrogen bond with D505. Extracellular proton may be attracted by the core domain and delivered to D505, then transferred through proton carriers including water or ammonia molecules, and finally released by E675. Molecules in the binding pocket of $BTR1_{OF/NH3}$ may be associated with the hydration of the pocket and could be utilized for proton transportation.

To investigate the effects of specific amino acids at the substrate binding site on BTR1 transport activity, a series of point mutations were introduced into BTR1 and expressed in HEK293F cells. The BTR1 mutants traffic to the cell membrane surface with little or

indistinguishable differences from wild-type BTR1-HA (Supplementary Fig. 4a, b). Furthermore, the electrophysiological experiments show that P723A, H724A and H719A-H724A mutations partially attenuate the $NH_3$ stimulated $H^+$ currents, especially at higher $NH_3$ concentration, while E675Q mutation almost eliminates the $NH_3$ stimulated $H^+$ currents (Fig. 2i). These observations confirm P723, H719, H724 and the negatively charged E675 contribute to BTR1's unique transport activity of proton which could be stimulated by $NH_3$.

## Interaction between $PIP_2$ and NTD is essential for the substrate transport process of BTR1

$PIP_2$ has been widely reported to function as an intracellular phosphoinositide regulator[29], which can regulate the activity of channels[30] and transporters[31,32], including the SLC4 family transporters NBCe1 (SLC4A4)[33,34] and NBCn1 (SLC4A7)[35]. Because of the high resolution of BTR1 structures, we were able to identify the density which clearly shows the molecular features of $PIP_2$ (Figs. 1c and 3b, c). $PIP_2$ molecules were derived from BTR1 expressing HEK293F cells. The two phospholipid tails of $PIP_2$ interact with the H5 helix and TM13 of the TMD through hydrophobic interactions (Supplementary Fig. 4c). Moreover, the inositol head of $PIP_2$ is surrounded by a positively charged pocket formed by TM13 of the TMD as well as by the α1 helix, the loop between the β3 and β4 sheets and the loop between the β5 sheet and the α4 helix of the NTD (Fig. 3a–c). We termed the involving charged residues R125, R128, R227, K260, K263, R827 and polar residue Q826 together as the lipid binding site (L site). These residues form extensive electrostatic interactions with the phosphate groups of $PIP_2$ inositol head (Fig. 3b, c). We next performed molecular dynamics simulations with the $BTR1_{OF/APO}$ state structure. During a 1 μs simulation, the overall structure of BTR1 gradually stabilizes (Supplementary Fig. 4d). In this process, $PIP_2$ still binds stably with BTR1, as the relative distances between the phosphate groups of $PIP_2$ and the adjacent amino acid side chains remain low (Supplementary Fig. 4e). We also calculated the binding free energy of one molecule of $PIP_2$ binding to BTR1 using MM/GBSA[36]. The value obtained is −100.66 kcal/mol that confirms the stability of $PIP_2$ binding. Structural alignment of the $BTR1_{OF/APO}$ state with AE1 (pdb 8CT3) by the gate domain shows that $PIP_2$ molecules bind to different regions of these transporters (Fig. 3d), which indicates that $PIP_2$ plays a different role in the entire transport processes of AE1 and BTR1.

As noted earlier, loss of function of BTR1 caused by the R125H mutation has been widely reported[9,37]. R125 is conserved among BTR1 orthologs but not among the SLC4 family members (Fig. 3e). Given that the binding of $PIP_2$ is pH sensitive and histidine is less positively charged than arginine under physiological pH conditions, we presume that the pathogenesis of R125H mutation can be attributed to the charge alteration of the L site which in turn adversely affects the binding of $PIP_2$, ultimately leading to the impaired transport activity of BTR1. We performed the mutations of R125, R128, R227, K260 and K263 in the L site and measured the $NH_4Cl$ stimulated currents, respectively. All these mutants traffic normally (Supplementary Fig. 4b). The electrophysiology experiments results confirm that the R125H mutation attenuates the transport activity of BTR1 (Fig. 3f), which is consistent with previous studies[6,37]. $NH_3$ stimulated currents of K260A and K263A mutants are also significantly reduced at extracellular pH (pHe) 7.4 and 8.0 (Fig. 3f). Previous studies have reported that the transport activity of BTR1 is enhanced by intracellular alkalization, but the enhancement is abolished by R125H mutation[38,39]. Thus, we hypothesize that $PIP_2$ molecules participate in the pH sensing process of BTR1 through passively changing their binding affinity with BTR1 under alkaline or acidic pH. Therefore, we measured the $NH_3$ stimulated currents of wild-type BTR1 and BTR1-R125H mutant at different intracellular pH (pHi) to test whether BTR1 with disrupted $PIP_2$ binding site is more sensitive to acidic pH. The electrophysiology experiment results show that the current densities of wild-type BTR1

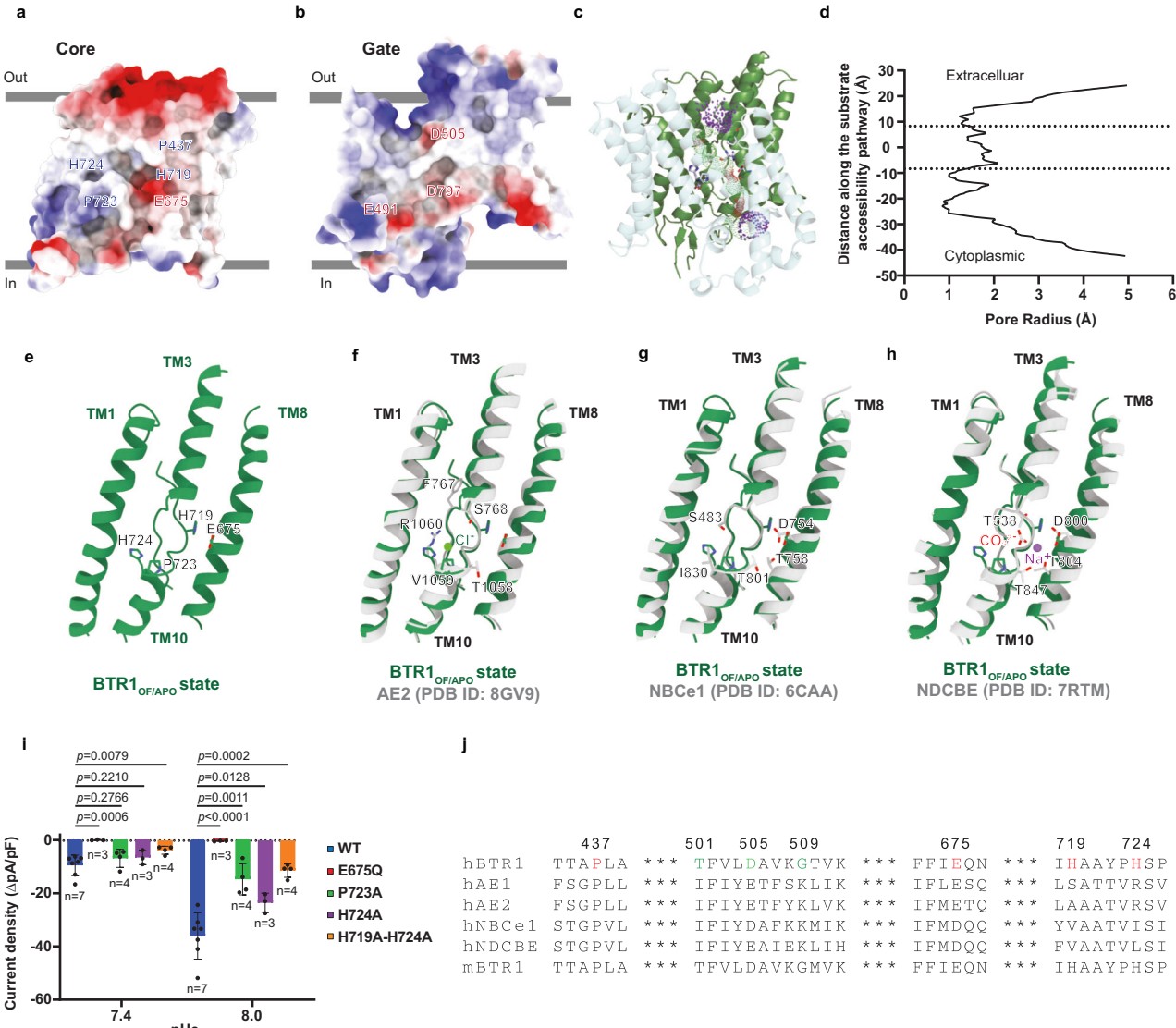

**Fig. 2 | Substrates binding sites of BTR1. a, b** Electrostatic surface representation of the core domain (**a**) and gate domain (**b**) of BTR1 in the outward-facing state. Functionally important amino acids are shown in blue or red according to their charge. **c, d** Ion permeation pathway of BTR1 in the outward-facing state conformation (**c**) and the pore radius values along the pathway (**d**). The dotted lines represent the extracellular and intracellular borders of the substrate binding site. The gate domain and core domain are colored in blue and green, respectively. The analyses were performed using HOLE2. **e** Substrate binding pocket of BTR1$_{OF/APO}$. The residues participating in the substrates coordination of BTR1 are labeled. **f–h** Alignment of the substrate binding pocket of BTR1$_{OF/APO}$ with AE2 (PDB ID: 8GV9) (**f**), NBCe1 (PDB ID: 6CAA) (**g**) and NDCBE (PDB ID: 7RTM) (**h**). The residues participating in the substrates coordination of AE2, NBCe1 and NDCBE are labeled.

The Cl$^-$ ion was shown as green sphere in (**f**). The CO$_3^{2-}$ ion and Na$^+$ ion were shown as stick model and purple sphere in (**h**), respectively. **i** Functional analysis of the residues involved in the substrate binding pocket formation. The current density values were calculated by the extreme difference of the current values after addition of 5 mM NH$_4$Cl at extracellular pH (pHe) 7.4 and 8.0. The current values are standardized by cell capacitance. Data shown are mean values ± s.d. of n biologically independent experiments and $p$-values were calculated by two-sided unpaired $t$-tests. **j** Sequence alignment of the residues in the pore region from *Homo sapiens* BTR1, *Homo sapiens* AE1, *Homo sapiens* AE2, *Homo sapiens* NBCe1, *Homo sapiens* NDCBE and *Mus musculus* BTR1 by PROMALS3D. Amino acids which are important for substrates coordination and permeation pathway formation are colored in red and green respectively.

hardly change as pHi dropped from 7.4 to 7.0. However, those of BTR1-R125H mutant are significantly reduced (Fig. 3g). This difference indicates that robust PIP$_2$ binding can enhance the tolerance of BTR1 to intracellular acidification, which is consistent with our hypothesis.

In summary, our data demonstrate that PIP$_2$ binds in the positively charged pocket formed by the TMD and NTD of BTR1 and plays essential roles in the complete substrate transport process of BTR1.

**BTR1 adopts inward-facing conformation in the absence of PIP$_2$**
To further explore how PIP$_2$ participates in the complete substrate transport process of BTR1 and prove the connection between PIP$_2$ and pH sensing, we attempted to solve the structures of BTR1 with the

R125H mutation or wild-type BTR1 prepared under acidic pH, which may both disrupt the interaction between BTR1 and PIP$_2$, and succeed in achieving resolutions of 2.96 Å and 2.94 Å, respectively (Supplementary Fig. 5). Intriguingly, two structures are nearly indistinguishable in conformation (RMSD = 0.250 Å) (Supplementary Fig. 6a), and both lack the densities of PIP$_2$. These results suggest that the disruption of PIP$_2$ binding site and acidic pH can both lead to the dissociation of PIP$_2$. Non-PIP$_2$ binding BTR1 occupies a 3D space of ~100 Å × 55 Å × 85 Å (Fig. 4a, b). Compared with the structures of BTR1 in the outward-facing state, the distinct features of the non-PIP$_2$ binding BTR1 structures involve that the NTD has undergone a significant twist, as well as an apparent conformational change has taken place in the

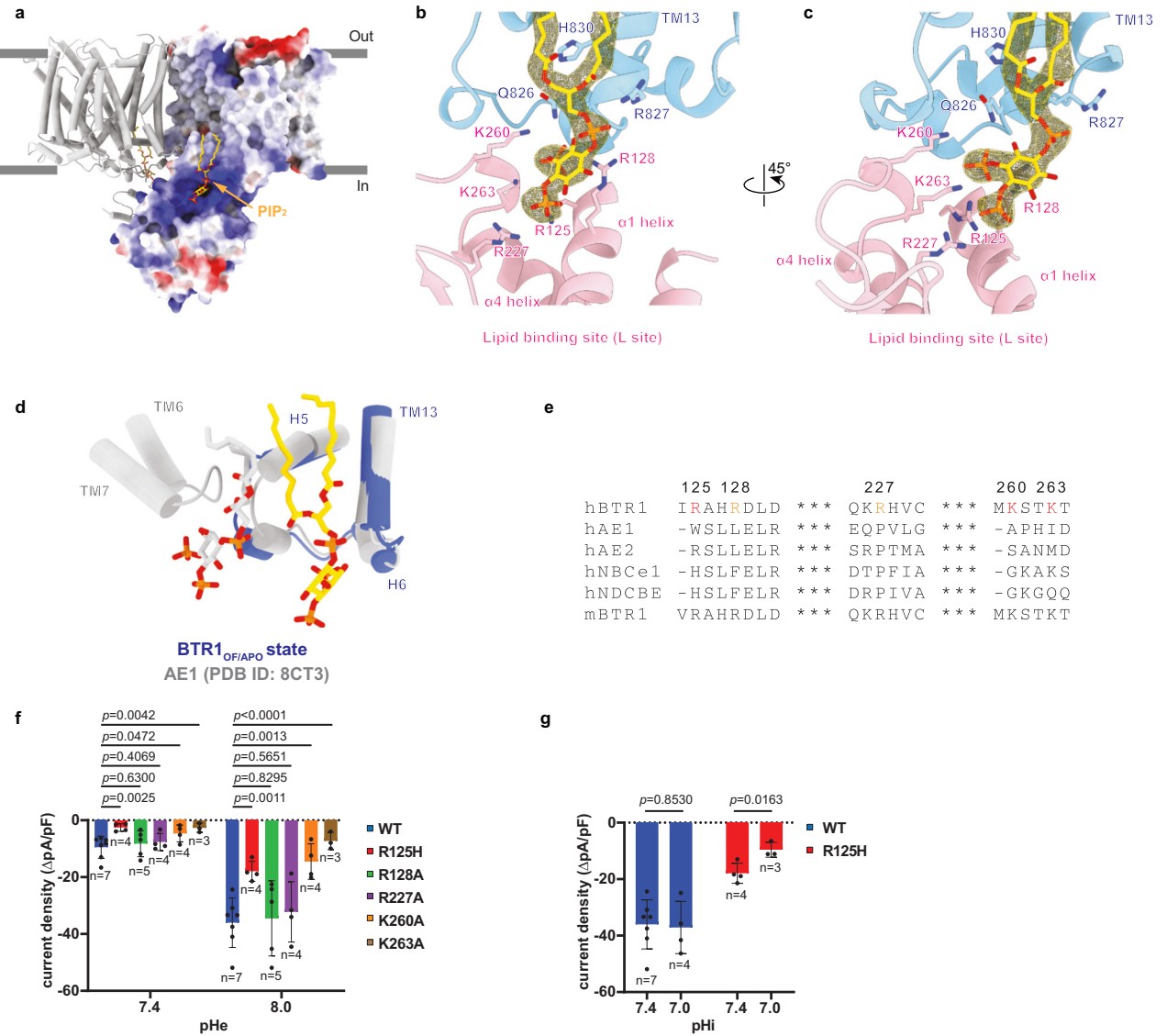

**Fig. 3 | PIP₂ binding site of BTR1 in the outward-facing state. a** $BTR1_{OF/APO}$ dimer with one monomer shown as electrostatic surface representation. $PIP_2$ molecules are shown as gold sticks. **b, c** Close-up views of the $PIP_2$ binding site. The $PIP_2$ density is shown in mesh. The $PIP_2$ molecule and its surrounding residues are shown in stick representation. TMD, NTD and $PIP_2$ are colored in blue, pink and gold, respectively. **d** Structural comparison of the $PIP_2$ binding sites of $BTR1_{OF/APO}$ (blue, with $PIP_2$ colored in gold) and AE1 (gray, with $PIP_2$ colored in gray) (PDB ID: 8CT3) aligned by the gate domains. Helices of BTR1 and AE1 are labeled in blue and gray, respectively. **e** Sequence alignment of the NTD residues forming the $PIP_2$ binding site from *Homo sapiens* BTR1, *Homo sapiens* AE1, *Homo sapiens* AE2, *Homo sapiens* NBCe1, *Homo sapiens* NDCBE and *Mus musculus* BTR1. Residues that are important for $PIP_2$ coordination and involved in forming the $PIP_2$ binding pocket (L site) are colored in red and orange, respectively. **f** Functional analysis of the residues in the $PIP_2$ binding site. The current density values of wild-type and mutant BTR1 proteins after the addition of 5 mM $NH_4Cl$ at pHe 7.4 and 8.0 as measured by whole-cell patch-clamp. Data shown are mean values ± s.d. of n biologically independent experiments and *p*-values were calculated by two-sided unpaired *t*-tests. **g** The current density values of wild-type BTR1 and BTR1-R125H proteins at different intracellular pH (pHi 7.0 and 7.4), measured by whole-cell patch-clamp. Data shown are mean values ± s.d. of n biologically independent experiments and *p*-values were calculated by two-sided unpaired *t*-tests.

TMD. Analysis of the TMDs using HOLE2 shows that the substrate binding sites of BTR1 in these two states are accessible from the intracellular side (Fig. 4c, d), suggesting their inward-facing conformation.

The structural alignment of the TMDs of $BTR1_{OF/APO}$ state and $BTR1_{IF/R125H}$ state by the gate domains shows that, during the conformation transition process from the outward-facing state to the inward-facing state, the gate domains remain relatively unchanged (RMSD = 1.678 Å), but an obvious displacement has occurred in the core domains (RMSD = 3.784 Å) (Fig. 4e, f and Supplementary Fig. 6e). In specific, the core domain of BTR1 has gone through a movement vertically towards the cytoplasm and horizontally towards the dimer interface (Fig. 4e, f). As for the gate domain, only H6 and H7 helices

shift towards the dimer interface to a certain extend (Fig. 4g). The core domain of $BTR1_{OF/APO}$ state needs to go through a 12.38° rotation and a 7.4 Å shift towards the gate domain to reach the position of core domain of $BTR1_{IF/R125H}$ state. The pore region formed by TM1, TM3, TM8 and TM10 has undergone similar movements. We define the centroid of P437 and P723 as the center of the pore region and observe that the centroid has moved 4.23 Å during the state transition process by aligning the gate domains (Fig. 4h). Alignment of the core domains and gate domains of $BTR1_{OF/APO}$ state and $BTR1_{IF/R125H}$ state separately show that the TMs arrangements within the core domain (RMSD = 1.093 Å) (Supplementary Fig. 6b) and the gate domain (RMSD = 0.547 Å) (Supplementary Fig. 6c) change slightly. These features are consistent with the previous studies which propose that SLC4 family

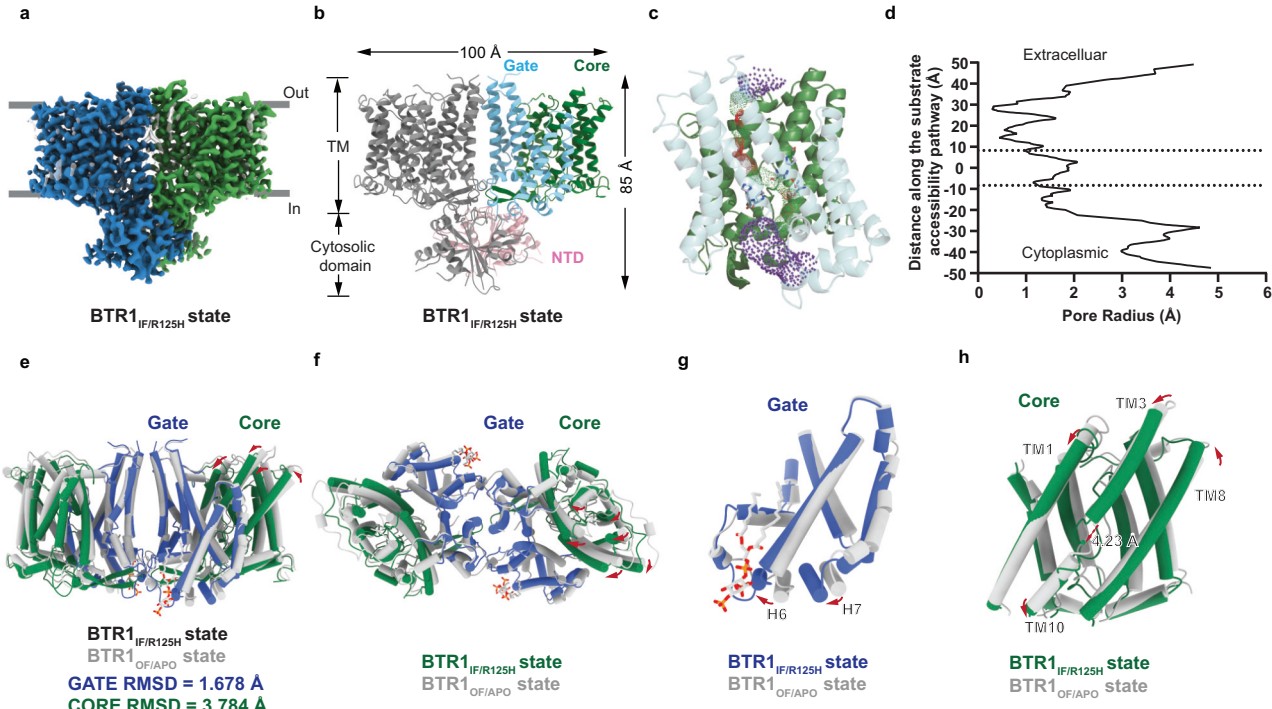

**Fig. 4 | TMDs of BTR1 in the inward-facing conformation. a, b** Cryo-EM map and structural model of the BTR1 dimer in the inward-facing state as viewed from the side. The color schemes are the same as in Fig. 1 (**c–e**). **c, d** Ion permeation pathway of BTR1 in the inward-facing state conformation (**c**) and the pore radius values along the pathway (**d**). **e, f** Structural alignment of TMDs of $BTR1_{IF/R125H}$ and $BTR1_{OF/APO}$ as viewed from the side (**e**) and the bottom (**f**). The shift directions of the transmembrane helices when transitioning from the outward-facing state to the inward-facing state are shown as red arrows. The alignment RMSDs of the gate

domains and core domains are labeled below. **g** Structural alignment of the gate domains of $BTR1_{IF/R125H}$ and $BTR1_{OF/APO}$. The helices with marked displacement are labeled in black. The shift direction of helices are shown as red arrows. **h** The centroid shift of the pore region during the state transition process. The centroids of residues P437 and P723 of $BTR1_{IF/R125H}$ and $BTR1_{OF/APO}$ are shown as green and gray spheres, respectively. The centroid shift is shown as red dotted arrow. The shift direction of helices are shown as red arrows.

transporters implement the elevator-like transport mechanism[20,40,41]. Similar displacements of the core domain have been observed in some transporters of other families which execute the elevator-like mode transportation, including NHE1 (-10° rotation of the core domain, 5 Å displacement of the Cα of D267)[42] and PIN8 (-20° rotation of the core domain, 5 Å displacement of the substrate binding site)[43]. These results support that BTR1 acts by the elevator mechanism.

### NTD of BTR1 in the inward-facing conformation

In comparison with the alteration of the TMD caused by the disruption of the interaction between $PIP_2$ and BTR1 during the state transition process, the conformation of the NTD changes even more significantly. The NTD has undergone a 175.9 ° deflection and a 24.9 Å position shift from $BTR1_{OF/APO}$ state to $BTR1_{IF/R125H}$ state conformation (Fig. 5a, b).

Intriguingly, the arrangement of the dimeric NTD of the $BTR1_{OF/APO}$ state is similar as that of the $BTR1_{IF/R125H}$ state with a RMSD of 1.848 Å (Fig. 5c). In the absence of $PIP_2$, residues which participate in the formation of the L site in the $BTR1_{OF/APO}$ state, have flipped to the cytosolic side of the whole frame (Fig. 5c). Densities of α1, the loop between α1 and β1 and the loop between α4 and β8 of BTR1, which can be clearly seen in the $BTR1_{OF/APO}$ state, disappear in the $BTR1_{IF/R125H}$ state due to the loss of interaction with $PIP_2$ and TMD (Supplementary Fig. 6d).

Meanwhile, the interface between the TMD and NTD of $BTR1_{IF/R125H}$ state has changed: the loop between β4 and β5 sheets, which is originally located at the bottom of the NTD and cannot be seen due to its flexibility in the $BTR1_{OF/APO}$ state, has flipped up to the top and interacts with H6 helix and the loop between H5 and H6 helices of the gate domain through hydrogen bonds (formed between V205 and Y820, Q212 and V739, E736 and S213) and extensive hydrophobic

interactions (Fig. 5d, e). The mutual repulsion between R209 of β5 (only stable in the inward-facing state) and R738 of 10β1 leads to the 10β1 and 10β2 sheets of the core domains being pushed away from the gate domains by β4 and β5 of the NTDs (Fig. 5d), which may trigger the overall horizontal shift of the core domains compared with the outward-facing state. In addition, the electrostatic interaction between R282 and E736 as well as the hydrogen bonds formed between the gate domain and the loop between β7 and β8 of the other NTD (between R822 and T145, T813 and P237, N239 and T813, N239 and A814) further stabilize the interactions between the TMDs and NTDs of BTR1 in the inward-facing conformation (Fig. 5f). R822 of H6 helix, which interacts with the NTD within a BTR1 monomer in the outward-facing state, turns to interact with the loop between α1 and α2 helixes (T146) in the NTD of the other monomer (Fig. 5f).

In short, the NTDs of $BTR1_{OF/APO}$ state turns nearly upside down due to loss of interaction with $PIP_2$, which further facilitate the transition of TMDs and establish new interactions with TMDs to stabilize BTR1 in the inward-facing conformation.

## Discussion

Despite BTR1 was first named as bicarbonate transporter-related protein-1 because of its sequence homology with the transporters of the SLC4 family, it does not transport bicarbonate like other family members. There is debate about whether $NH_3$ co-transport with $H^+$ or $NH_3$ activate $H^+$ transport activity. High-resolution maps of $BTR1_{OF/APO}$ state and $BTR1_{OF/NH3}$ state allow us to unambiguously identify the side chains of residues in the substrate coordination site. The mutagenesis analysis results show that H719, H724 and P723, which are not conserved among the SLC4 family (Fig. 2e–h, j), participate in the substrate transport process and play important roles in the presence of high

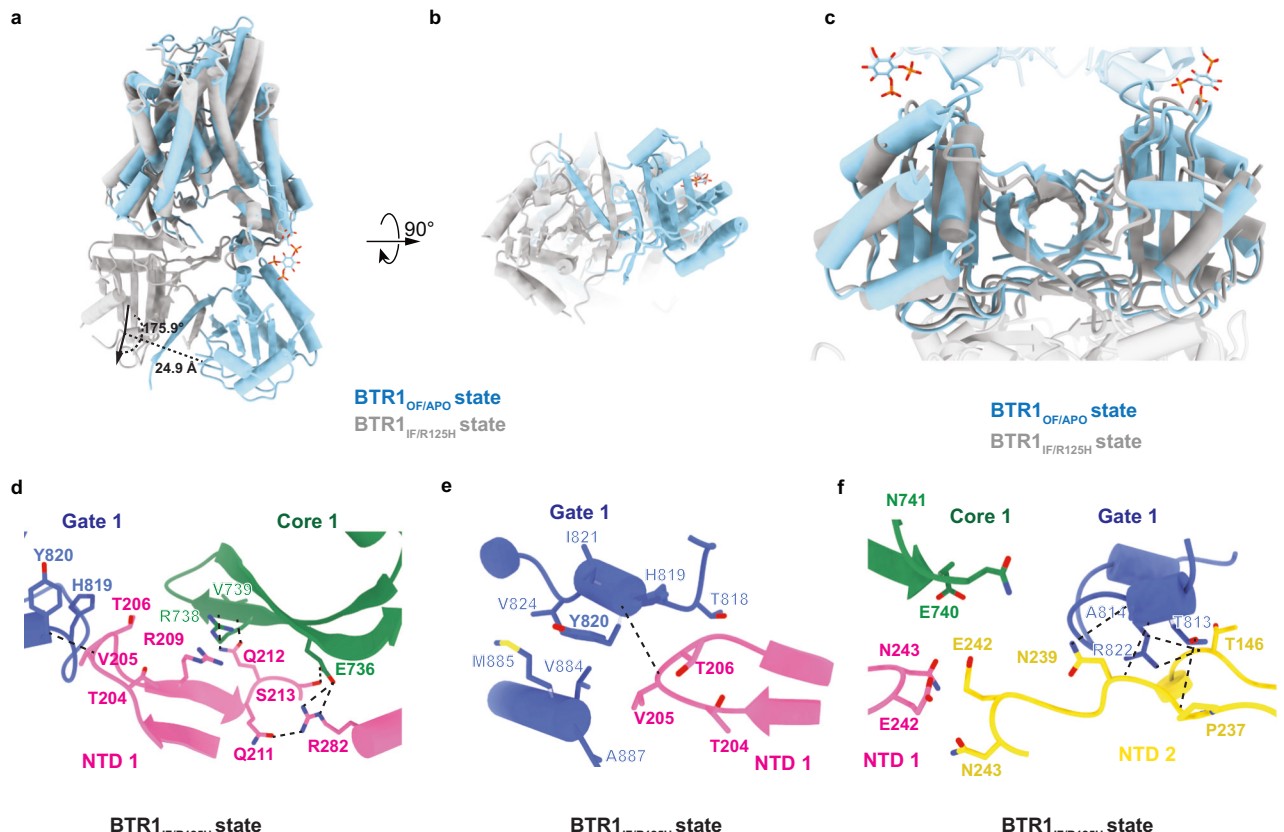

**Fig. 5 | NTDs of BTR1 in the inward-facing conformation. a, b** Structural comparison of BTR1$_{OF/APO}$ (blue) and BTR1$_{IF/R125H}$ (gray) monomers aligned by their TMDs viewed from the side (**a**) and the bottom (**b**). The rotation angle and centroid shift between BTR1$_{OF/APO}$ and BTR1$_{IF/R125H}$ are marked with dotted lines. **c** Structural comparison of BTR1$_{OF/APO}$ (blue) and BTR1$_{IF/R125H}$ (gray) aligned by their NTDs. The TMDs are both shown in transparent mode. **d** Hydrophobic interactions and hydrogen bonds formed between the NTD and TMD of one

BTR1$_{IF/R125H}$ monomer. The NTD, gate domain and core domain of one BTR1$_{IF/R125H}$ monomer are colored in pink, blue and green, respectively. The interactions are shown as dotted lines. **e** Close-up view of the hydrophobic interactions between the NTD and gate domain of one BTR1$_{IF/R125H}$ monomer. The hydrophobic interaction between V205 and Y820 is shown as dotted lines. **f** Hydrogen bonds between the gate domain of one BTR1$_{IF/R125H}$ monomer and the NTD of the other monomer. The interactions are shown as dotted lines.

concentration of NH$_3$ (Fig. 2i). E675Q mutation totally abolishes the H$^+$ currents (Fig. 2i), which is consistent with the influences of the mutant counterparts of other SLC4 family transporters and can explain the pathogenic mechanism of this mutation. Though our structures cannot settle the dispute in this field, the densities along the substrate permeation pathway reveal the importance of the surrounding residues. We have also provided precise structure models for further functional studies and MD simulations.

Previous studies have reported that PIP$_2$ participate in the regulation of the transport activity of SLC4A4 and co-localize with SLC4A1[33,44,45]. Recently solved cryo-EM structures of prestin and band 3–ankyrin multi-protein complex are with the combination of cholesterol or PIP$_2$[19,46]. Prestin and AE1 are structurally similar to BTR1. The observed combination of cholesterol with the TMDs of prestin provides direct evidence that lipid can act as a regulatory element and play an essential role in the transporter function[46]. The combined PIP$_2$ in AE1 may participate, at least partially, in the recruitment of protein 4.2[19]. Our structure of BTR1$_{OF/APO}$ state is also in complex with PIP$_2$. Disruption of the PIP$_2$ binding sites (BTR1$_{IF/R125H}$ state) or acidic pH (BTR1$_{IF/5.5}$ state) both turn BTR1 into the inward-facing conformation. The conformational changes induced by the adjustment of pH have also been observed in recently solved AE2 structures[21]. The mutagenesis analysis prove the importance of R125H, K260A and K263A mutations in BTR1 substrates transport activity and BTR1 R125H mutant is more sensitive to acidic pH than wild-type BTR1. Previous studies have proposed that NH$_3$ activate BTR1 by inducing the alkalization of the intracellular environment[39], which can be achieved

through the transport of BTR1 or diffusion across the cytoplasmic membrane. Here we suggest that PIP$_2$ may participate in sensing the intracellular pH by changing its affinity to BTR1. BTR1 with a disrupted PIP$_2$ binding site needs more alkaline pH to interact with PIP$_2$ to accomplish substrates transportation.

The transport mechanisms of transporters are divided into three categories: rocker switch, rocker bundle and elevator[47]. The transport process of the SLC4 family transporters has been widely considered as an elevator mode and the recently solved structures of AE1 in both inward-facing and outward-facing state further support the mechanism[20,25,40,41]. Through structural analysis, we also believe that BTR1 works in this mode and we propose the following model (Fig. 6). In the apo state, BTR1 binds to PIP$_2$ in a stable manner (Fig. 6a). PIP$_2$ molecules then dissociate from the NTD of BTR1 after ions reach the substrate binding sites, leading to the conformation of BTR1 changing into an inward-facing state or a state between inward-facing and outward-facing (occluded state) (Fig. 6b). Protonation of PIP$_2$ under acidic pathological condition or pathogenic mutations exist in the L site of BTR1 lead to the disruption of the interaction between PIP$_2$ and BTR1. The NTD dimer then turns upside-down and inserts β4 and β5 sheets into the gate domains and finally stabilizes BTR1 in the inward-facing conformation (Fig. 6c). Our model is in agreement with the recently solved structures of AE2 in different states regulated by pH[21] and supplement the pH-sensing mechanism that regulatory ligands, like PIP$_2$, might cooperate with the NTD in the regulation of SLC4 family activity.

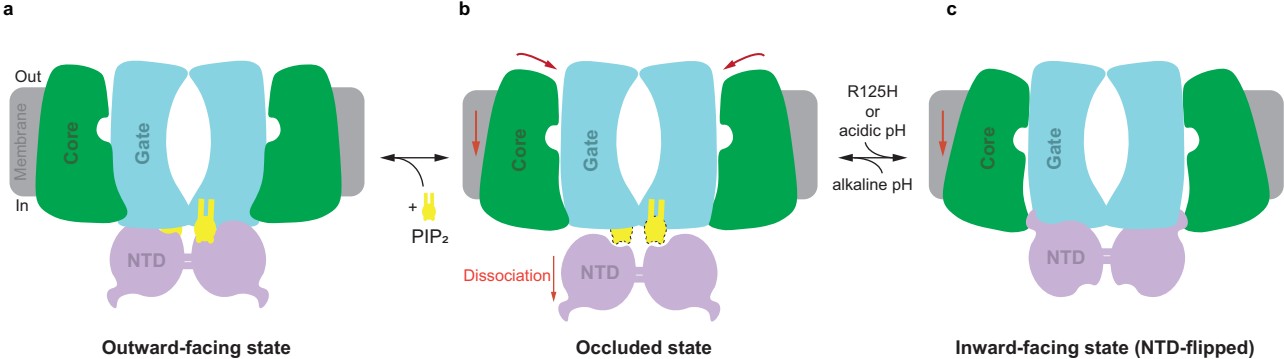

**Fig. 6 | BTR1 state transition process. a** BTR1 is in the outward-facing state with $PIP_2$ stably binding. **b** Dissociation of $PIP_2$ molecules from the NTDs of BTR1 after ions reach the substrate binding sites induces the rotation of the core domains, leading to the occluded state of BTR1. **c** Disruption of the interaction between $PIP_2$ and BTR1 due to acidic pathological condition or pathogenic mutation in the L site, triggers the flip of the NTD and the conformational change into the inward-facing state.

In summary, we present the cryo-EM structures of full-length BTR1 in the outward-facing state in complex with $PIP_2$ and in the inward-facing state under pathological conditions. These structures shed light on the structural basis of how BTR1 implement its transport activities. Our experimental results also provide important insights into understanding how $PIP_2$ participate in the conformational transition, pH sensing and functional regulation of BTR1.

## Methods

### Cell culture
HEK293F cells (Thermo Fisher Scientific) were cultured in suspension in Freestyle 293 medium (Gibco) supplemented with 1% fetal bovine serum (FBS) at 37 °C with 6% $CO_2$ and 70% humidity.

### Constructs
Full-length *SLC4A11* gene (UniProtKB accession: Q8NBS3) was amplified from human HEK293 cDNA and then cloned into PCGFP-BacMam (PBM) plasmid (kindly provided by Eric Gouaux lab[24]) using EcoR1 and Xho1 restriction sites. A series of point mutations were carried out with the PBM-*SLC4A11* plasmid by the KOD-PLUS-neo enzyme (TOYOBO). To measure the plasma membrane expression of BTR1 and its mutants, we inserted a HA-tag (YPYDVPDYA) into the PBM-*SLC4A11* plasmid between aa 544 and aa 545, which is located at a flexible loop in the extracellular region of the transporter.

### Expression and purification
BTR1 wild-type and R125H mutant proteins were overexpressed in HEK293F cells using the Bac-to-Bac baculovirus expression system (Invitrogen, USA). The PBM plasmids were transformed into DH10B cells to produce bacmid and then SF9 insect cells were used to mediate the package and amplification of Baculovirus. Subsequently, a 1:10 ratio of baculovirus against 293F cells was used to infect HEK293F cells and the cells were incubated at 37 °C in suspension supplemented with 1% (v/v) FBS and 5% $CO_2$ in a shaking incubator. Twelve hours later, 10 mM sodium butyrate was added to the culture. The cells were incubated for further 48 h before harvesting.

The HEK293F cells expressing BTR1 proteins (wild-type or R125H) were collected by centrifugation at 4000 g for 10 min and resuspended with 1 × TBS buffer (140 mM NaCl, 3 mM KCl and 30 mM Tris-HCl, pH 7.4). The resuspended cells were ultrasonically extracted after the addition of 10 mM PMSF and then centrifuged at 8000 g for 20 min to remove the cell nuclei and organelles. The supernatant was removed and centrifuged at 100,000 g for 1 h to collect the cell membrane. The precipitate was solubilized with dissolving buffer (150 mM NaCl, 20 mM Tris-HCl pH 7.4, 1% (w/v) n-dodecyl β-D-maltoside (DDM, Anatrace), and 0.2% (w/v) cholesteryl

hemisuccinate (CHS, Anatrace)), and was then stirred at 100 g at 4 °C for 1 h. The mix was centrifuged at 100,000 g for 30 min and then the supernatant was collected and mixed slowly with Ni-NTA beads. Non-specifically bound protein was washed using 10 columns of purification buffer (150 mM NaCl, 20 mM HEPES, pH 7.4 or pH 5.5, and 0.01% (w/v) glyco-diosgenin (GDN, Anatrace)) in addition with 20 mM or 40 mM imidazole. The target protein was eluted using 2 columns of purification buffer in addition to 250 mM imidazole. GFP and His-tag of the eluted protein were removed by incubating with TEV protease at 4 °C overnight, and then concentrated and further purified by gel filtration (Superose 6 Increase 10/300 GL, GE Healthcare, USA). Peak fractions were collected and concentrated to nearly 10 mg/mL for cryo-EM sample preparation.

### Cryo-EM sample preparation and data acquisition
The concentrated protein samples were loaded onto glow-discharged quantifoil R 1.2/1.3 200 Holey Carbon films Au 200 mesh grids and blotted with filter paper for 3 s under 100% humidity at 4 °C before being plunged into liquid ethane with a FEI Vitrobot Mark IV. A 300 kV Titan Krios electron microscopy with a Gatan K2 Summit direct electron detection camera was used for data collection. A calibrated magnification of 165kx with a pixel size of 0.821 Å was used for acquisition of movies. 8 e/pixel/s dose rate and total dose of 52.2 e/Å² was used. Each 4.4 s movie was dose-fractioned into 40 frames. For each protein condition, >2000 micrographs were collected for structural analysis.

### Image processing
We used the cryoSPARC software for structural analysis[48]. The original movies were aligned and motion corrected with Patch motion correction step. The averaged micrographs were generated by 2× binning. The corrected micrographs were carried out with Patch CTF estimation for CTF measuring and further image selection. Bad images with inappropriate power spectra and ice thickness were deleted within the Manually Curate Exposures step. Protein particles were initially picked using Blob picker with a proper box size and then extracted from micrographs. The extracted particles were then subjected to 2D classification to generate 100 2D classes. Nearly 10–20 good 2D classes were selected to generate several initial models using the Ab-initio reconstruction step. The best initial model was used to create templates and the templates of different angles were further used for template picking. After a second round of particle extraction, 2D classification and 2D class selection, a new initial model was generated for further 3D reconstruction and refinement. Non-uniform refinement was performed to enhance the resolution. Ultimately, local refinement was carried out to optimize the 3D map to the highest resolution. We

used the 0.143 Å criterion of 'gold-standard' Fourier shell correlations (FSCs) to estimate the overall resolution within cryoSPARC. Local-resolution estimation of reconstructed maps were also determined within cryoSPARC[48].

## Model building and refinement

We used alphafold[49] to build the initial models of the NTD and TMD of BTR1 and merged them in Coot[50]. The two BTR1 domains of BTR1 are similar to those of other SLC4 family members, especially TMD, as the substrate binding site is in nearly the same position. The N-terminal 1–103 residues, the flexible regions between the NTD and the TMD (reside 308 to reside 335) and the C-terminal 888–891 residues could not be built due to their flexibility. PIP$_2$ was built with TMD based on the density of map. We used the real-space refinement mode of phenix[51] to further refine the model.

The inward-facing state of BTR1 was built as described above. Because of insufficient density, some residues are still missing. In view of the rotation of NTD in this state, we used a different initial model. PIP$_2$ was not built because the density disappeared in these two models, but the rest of the steps are the same as above.

## Assessment of membrane protein expression

To assess the membrane expression of the constructs encoding BTR1 and its mutants, we inserted a HA-tag (YPYDVPDYA) into the PBM-*SLC4A11* wild-type and mutant plasmids between aa 544 and aa 545, which is located on a flexible extracellular loop of the transporter. Wild-type *SLC4A11* and its mutants were transfected into HEK293F cells with polyethylenimine (PEI) (Polysciences) at a cell density of $1.5 \times 10^6$/ml. Cells were harvested 36 h after transfection. Approximately $2 \times 10^6$/ml of each cell type were collected by centrifugation at 1000 g for 1 min and washed with PBS buffer twice. Cells were then incubated with 50 µl PBS with 1% FBS (PAN Biotech) and 0.1 µl PE anti-HA.11 Epitope Tag Flow Cytometry Antibody (BioLegend) at 37 °C for 30 min. Cells were washed twice in PBS and resuspended in 500 ul PBS, and then transferred into 5 ml polystyrene round-bottom tubes for flow cytometry analysis. FITC and PE channel were detected sequentially, representing the target protein total expression (GFP) and membrane expression (PE-conjugated anti-HA), respectively. The ratio of cell numbers and fluorescence intensity of the two signals were calculated within FlowJo v10.6.2.

## Electrophysiology

BTR1 constructs were transfected into FreeStyle 293 F cells using PEI at a cell density of $1 \times 10^6$ cells/ml. Cells were cultured in FreeStyle 293 Expression Medium with 1% FBS (PAN Biotech) for 24–36 h before making recordings. Currents were recorded using whole-cell mode at −55 mV or recorded at a holding potential of −55 mV and a series of 400-ms voltage pulses with 15-mV increments (−100–+50 mV) through an Axon-patch 200B amplifier (Axon Instruments, USA). Patch electrodes were pulled by a horizontal micro-electrode puller (P-1000, Sutter Instrument Co, USA) to a tip resistance of 1.0–3.0 MΩ. Pipette solution [containing (mM): 135 cesium gluconate (CsOH mixed with gluconic acid), 1 CaCl$_2$, 10 tetraethylammonium chloride (TEA-Cl), 10 EGTA, and 10 HEPES (pH 7.4, CsOH)] and bath solution [containing (mM): 140 TMACl, (or 135 TMACl and 5 NH$_4$Cl) 1.5 CaCl$_2$, 10 CsCl, and 10 HEPES (pH 7.4, or 6.5, 8.0, HCl)] were used for recording steady-state currents and measuring the activating effect of NH$_3$. Pipette solution pH was changed to 7.0 when measuring H+ currents at different pHi. Signals were acquired at 5 kHz and low-pass filtered at 300 Hz. Data were further analyzed with pclampfit 10.0 software.

## Computational modeling

To gain deeper insights into the binding mechanism of PIP$_2$ to SLC4A11, we conducted in silico calculations. The configurations of PIP$_2$ were optimized at B3LYP/6−31 G* level using Gaussian 09 package. The starting configurations of the SLC4A11 and PIP$_2$ system for molecular dynamics (MD) simulation were attained from the cryo-EM structures. The complex was embedded in a POPC bilayer (190 and 174 POPC molecules in the upper and lower leaflet, respectively) to generate a suitable membrane system with 15 Å layers of water on both sides of the membrane and 150 mM NaCl using CHARMM-GUI[52].

MD simulation on SLC4A11/PIP$_2$ complex was performed with AMBER20[53] using AMBER ff99SB-ILDN, Lipid14 force field and Generalized Amber Force Field (GAFF). The topological parameters of PIP$_2$ were generated using RESP charge fitting in Antechamber. The particle mesh Ewald (PME) algorithm[36] was employed to compute long-range electrostatic energies, and van der Waals and Coulomb interactions were truncated at 10 Å. All hydrogen-related covalent bonds were constrained using the SHAKE algorithm.

The system firstly underwent minimization of side chains of the protein and PIP$_2$, respectively. 2500 steps of steepest-descent and 2500 steps of conjugate-gradient minimization then followed to remove unfavorable contacts. The system was gradually heated from 0 K to 298 K and then equilibrated for 500 ps at 1 atm in an isochoric/isothermal (NPT) ensemble with periodic boundary conditions. Temperature and pressure controls were achieved by Nosé-Hoover thermostat[54] and Berendsen barostat[55] with a frequency of 2.0 ps, respectively. Finally, the equilibrated systems were subjected to 1 µs-long production. All MD simulation processes were independently performed three times.

## Calculation of binding free energies

The Molecular Mechanics Generalized Born Surface Area (MM/GBSA) approach[36] has been successfully applied to predict the binding free energy ($\Delta G_{bind}$) in various protein-ligand complexes. Here, 1000 snapshots were extracted from the last 50 ns MD trajectory. The detailed calculations can be expressed as follows:

$$\triangle G_{bind} = \triangle H - T\triangle S \approx \triangle E_{gas} + \triangle G_{sol} - T\triangle S \qquad (1)$$

$$\triangle E_{gas} = \triangle E_{ele} + \triangle E_{vdW} \qquad (2)$$

$$\triangle G_{sol} = \triangle G_{pol} + \triangle G_{np} \qquad (3)$$

Where $\Delta E_{gas}$, $\Delta G_{sol}$ and $-T\Delta S$ represent the changes of binding energy in the gas phase, solvation and the conformational entropy upon binding, respectively. $\Delta E_{gas}$ includes $\Delta E_{ele}$ (electrostatic) + $\Delta E_{vdW}$ (van der Waals) potential. $\Delta G_{sol}$ contains contributions of polar ($\Delta G_{pol}$) and nonpolar ($\Delta G_{np}$) terms. Given that the binding conformational entropy ($\Delta S$) is computationally expensive and of limited accuracy, we assumed that $\Delta G_{bind}$ is approximately equal to the sum of $\Delta E_{gas}$ and $\Delta G_{sol}$.

## Reporting summary

Further information on research design is available in the Nature Portfolio Reporting Summary linked to this article.

## Data availability

The data that support this study are available from the corresponding authors upon request. The cryo-EM maps have been deposited in the Electron Microscopy Data Bank (EMDB) under accession codes EMD-32942 (BTR1$_{OF/APO}$), EMD-32943 (BTR1$_{OF/NH3}$), EMD-32941 (BTR1$_{IF/R125H}$), and EMD-32940 (BTR1$_{IF/5.5}$). Atomic model coordinates have been deposited in the Protein Data Bank (PDB) under accession codes 7X1I (BTR1$_{OF/APO}$), 7X1J (BTR1$_{OF/NH3}$), 7X1H (BTR1$_{IF/R125H}$), and 7X1G (BTR1$_{IF/5.5}$). Source data are provided with this paper.

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

## Acknowledgements

Cryo-EM data collection was supported by the Electron Microscopy Laboratory and Cryo-EM Platform of Peking University with the assistance of Xuemei Li, Zhenxi Guo, Xia Pei, Bo Shao and Guopeng Wang. P.Z. received training in cryo-EM sample preparation and data collection from Dr. Y. Mao from the School of Physics, Peking University. The paper was kindly modified by Dr. Susan Cole. This study is supported by grants to Y.Y. including National Natural Science Foundation of China (grant 82030081), the National Key Research and Development Program of China (2021YFA1300601); Shenzhen High-level Hospital Construction Fund and Shenzhen Basic Research Key Project (JCYJ20220818102811024).

## Author contributions

Y.Y. initiated and guided the project. Y.L. screened constructs and purified protein with assistance from P.Z., H.C., H.S., W.W., and Z.D.; P.Z. prepared cryo-EM samples and collected cryo-EM data with assistance of D.D. and Y.L.; Y.L. processed the cryo-EM data; Y.L. built and refined the atomic models with help from D.D. and L.L.; Y.L. and P.Z. performed the whole-cell electrophysiology with assistance of D.D.; P.Z. finished the assessment of protein membrane expression; P.Z. performed the data analysis and figure drawing with assistance of D.D.; H.X. and Y.C. performed the MD simulations; Y.J. supervised the experiments. The manuscript was written by P.Z., Y.L., D.D., and Y.Y.

## Competing interests

The authors declare no competing interests.
