## [Peer Review File · Nature Communications]

Structural insights into the conformational changes of BTR1/
SLC4A11 in complex with PIP2Reviewers' Comments:

Reviewer #1:

Remarks to the Author:

Overall comments:

SLC4A11 is an idiosyncratic member of the SLC4 family as it is the only SLC4 member not to transport bicarbonate and is the most evolutionarily distantly related from the other 9 members of the SLC4 family. SLC4A11 was originally described as a borate transporter before subsequent studies showed it to transport H⁺/OH⁻ and ammonia. Mutations in SLC4A11 are the molecular basis of multiple diseases, and a better understanding of SLC4A11 structure and function could be highly valuable. Moreover, understanding how lipids regulate transporter structure and function remains highly valuable not only for SLC4 proteins but for transporters in general. This work reports cryoEM structures of human SLC4A11 in inward- and outward-facing states with ligands including ammonia and PIP2. Provocatively, the authors claim that the loss or disruption of PIP2 binding leads to an inward-facing state. The structural data are of high quality, with quality maps shown and strong stats on their built models. These structural data have the potential to provide valuable new insights to the field. However, I have questions about the analysis of the data that I think the authors need to address. I have outlined my comments as either major or minor and present them below:

Major comments:

Line 70: Zhekova et al (PMID: 36517642) must be cited (for example in lines 70-71,) as it was the first paper to report both the inward- and outward-facing structures of the same SLC4 protein (bovine AE1.) Likewise, the paper reporting the inward- and outward-facing structures of AE2 (PMID: 37002221) also should be cited here. Their insights should be referenced here early, and also included in the discussion later.

Line 135: Be more precise and specific than stating the density for NH₃ is "strong." By what basis are you concluding it is ammonia rather than water or some other ion? Also, more must be said about where the NH₃ binding is observed to occur and how that compares with expectations. Obtaining substrate-bound SLC4 structures has been difficult, and one of few manuscripts to do so is this one from Wacker and colleagues: <https://www.biorxiv.org/content/10.1101/2022.02.11.480130v1>. How does the ammonia binding you observe compare to the bicarbonate binding seen there?

Figure 2: It would be helpful to have a view of the Core as in panels e-g, but with the numbering of SLC4A11 shown. This would help greatly for interpreting the data in panel h that uses the SLC4A11 numbering.

Line 168: What is the basis by which the PIP2 binding is described as "extremely tight?" The language that follows provides only a qualitative description of where the PIP2 is located and its protein contacts. Describing it as "extremely tight" suggests a numerical precision that is not justified here. The language needs to be defended or, more likely, changed if such evidence cannot be stated.

Line 199: It is stated here that, "Previous studies reported that transport activity of BTR1 is enhanced by intracellular alkalization." However, in line 205 it is stated that, "Electrophysiology experiment results showed that the current densities of wild-type BTR1 hardly changed when pHi dropped from 7.4 to 7.0." Although I agree that the subsequent data showing a current density for the R125H is significant, I see the two lines I quoted as a contradiction that is not acknowledged. What do the authors think is happening there?

Also, while the R125H data is nice, histidine can still retain a positive charge in near-neutral pH. Wouldn't it be reasonable to hypothesize that there would be an even more drastic (and pH-independent) difference when testing an R125A mutant? Did the authors ever construct such a

mutant, and if so what did it show? If the authors never made such a mutant, then why not?

Line 247: It should be noted here with appropriate citations that other SLC4 transporters have been proposed to act by the elevator mechanism. There is also an opportunity you should take here to compare the 4.23 angstrom shift you observe with those seen for elevator mechanisms. While yours is on the smaller magnitude (compared to that of elevator transporters like GltPh), there are other proposed elevator mechanisms that have similar shifts and noting that context would be useful here.

Line 317: More care must be taken to choose correct and proper citations. Arakawa et al (ref 17) did not propose SLC4A1 to use an elevator transport mechanism (the word "elevator" does not appear in that paper nor is such a mechanism described.) The first paper to report an elevator mechanism for SLC4 transporters and their homologs was Thurtle-Schmidt and Stroud (PMID: 27601653) and subsequently supported by Forrest and colleagues (PMID: 29167180) and Kurtz and colleagues (PMID: 36517642). The Arakawa reference should be removed from this line and the three references above should be inserted in their place.

Minor comments:

The title does not read grammatically correct and needs to be changed. In particular, "with PIP2 and conformational change" has nonparallel structure.

There are numerous word choices that need to be fixed for grammar or readability. For example line 27 should be "activation" not "activated." In lines 148-149 "which is likely account for" is incorrect word choice. There are many instances like these in the manuscript which will require close inspection to correct. Additionally, check for spelling throughout the entire paper. Words like "pipette" are misspelled.

Line 62: The phrase, "which severely reduces the quality of life of affected individuals" is not needed and should be deleted. Some might find that language ableist.

Line 74: structures do not "fail;" rather simply state that they "did not resolve..."

Line 378: state that the purpose of the TEV protease was, presumably, to remove the GFP and His-tag from the protein (this is never stated anywhere.)

Line 423: Be more precise than stating "some residues are still missing." Which residues are absent from the model?

The formatting of citation 12 has degree information included and needs to be fixed.

Reviewer #2:

Remarks to the Author:

This manuscript reports cryo-EM structures of human SLC4A11 transporter in several different conditions. The structures appeared to be of high quality, with resolutions ~3.0 angstrom. The structures revealed several interesting aspects about this family of transporters, including identification of a presumed PIP2 binding site, drastic conformational changes in the cytoplasmic domain, and potentially distinct conformations along the transport cycle. However, the structural and functional analysis and interpretation are not well presented. Overall, the manuscript would need significant improvement in terms of scientific accuracy as well as editing to improve grammar and readability. Below are some major points.

1. Page 6-7, substrate binding sites. The authors claimed densities corresponding to NH3 molecules in Supplementary Fig. 3h and 3i. A comparison of the cryo-EM densities in this region in the apo and

NH₄Cl structures would be important.

2. PIP₂ binding and functional consequence. Several basic residues, such as R128 and R227, have been identified to coordinate the PIP₂ headgroup but mutations R128A and R227A showed activity similar to the wild type (Fig. 3f). What is the explanation?

3. On page 11, structural comparison of BTR1 (outward-facing/APO) and BTR1 (inward-facing/R125H). The authors aligned these two structures by the 'gate' domain and then claimed that the 'gate' domain remains relatively unchanged (RMSD 1.678 Å) and the 'core' domain undergoes apparent changes (RMSD 3.784 Å). This is problematic. What happens if the 'core' domain is used to align these two structures? Are the 'gate' domain and the 'core' domain undergoing rigid-body motions? More structural analysis, such as comparison of domain movements within a protomer and between dimers, would help understand structural rearrangements in these two conformations.

4. Assignment of outward and inward facing conformations. From Figs. 2d and 4d, the openings at the cytoplasmic side for the assigned outward- and inward-facing conformations are similar. How do the authors assign the functional states of these two conformations to outward- and inward- facing, respectively?

5. Elevator mechanism. From structural comparisons and Fig. 6, it appears that the 'core' domain rocks against the 'gate' domain to generate alternate access. This would not fit with an elevator type of transport.

6. It seems that the complete transport pathway is within each subunit of the dimeric transporter. Do mutations at the dimer interface (the 'pore' domain and NTD) alter transport function?

We are grateful for the constructive suggestions from editors and reviewers. Please find the point-to-point responses below.

Reviewer #1 (Remarks to the Author):

Overall comments:

SLC4A11 is an idiosyncratic member of the SLC4 family as it is the only SLC4 member not to transport bicarbonate and is the most evolutionarily distantly related from the other 9 members of the SLC4 family. SLC4A11 was originally described as a borate transporter before subsequent studies showed it to transport H⁺/OH⁻ and ammonia. Mutations in SLC4A11 are the molecular basis of multiple diseases, and a better understanding of SLC4A11 structure and function could be highly valuable. Moreover, understanding how lipids regulate transporter structure and function remains highly valuable not only for SLC4 proteins but for transporters in general. This work reports cryoEM structures of human SLC4A11 in inward- and outward-facing states with ligands including ammonia and PIP2. Provocatively, the authors claim that the loss or disruption of PIP2 binding leads to an inward-facing state. The structural data are of high quality, with quality maps shown and strong stats on their built models. These structural data have the potential to provide valuable new insights to the field. However, I have questions about the analysis of the data that I think the authors need to address. I have outlined my comments as either major or minor and present them below:

Major comments:

Line 70: Zhekova et al (PMID: 36517642) must be cited (for example in lines 70-71,) as it was the first paper to report both the inward- and outward-facing structures of the same SLC4 protein (bovine AE1.) Likewise, the paper reporting the inward- and outward-facing structures of AE2 (PMID: 37002221) also should be cited here. Their insights should be referenced here early, and also included in the discussion later.

Response: We have cited these papers and referenced their insights in line 69-74, line 325-327 and line 337-340 as suggested.

Line 135: Be more precise and specific than stating the density for NH₃ is “strong.” By what basis are you concluding it is ammonia rather than water or some other ion? Also, more must be said about where the NH₃ binding is observed to occur and how that compares with expectations. Obtaining substrate-bound SLC4 structures has been difficult, and one of few manuscripts to do so is this one from Wacker and colleagues: <https://www.biorxiv.org/content/10.1101/2022.02.11.480130v1>. How does the ammonia binding you observe compare to the bicarbonate binding seen there?

Response: The cryo-EM densities of putative water molecules in BTR1_{OF/APO} states are shown in Supplementary Figure. 3j for better visualization of the differences from BTR1_{OF/NH3} states. Densities distributed along the TM5 and the substrate binding pocket are only putative H₂O and NH₃ molecules. Figure. 2f and 2h were revised to show the interaction between substrates and the core domains of SLC4 family proteins. Probable interactions between core domain of BTR1 and putative NH₃ molecules were discussed in line 152-159 as suggested by the reviewer. The bicarbonate binding site in AE1 solved by Wacker and colleagues is close to the Am1 binding sites, located within 4 Å to H724 (corresponding to R730 in AE1).

Figure 2: It would be helpful to have a view of the Core as in panels e-g, but with the numbering of SLC4A11 shown. This would help greatly for interpreting the data in panel h that uses the SLC4A11 numbering.

Response: We have provided revised Fig. 2e for better visualization of the BTR1 core domain.

Line 168: What is the basis by which the PIP₂ binding is described as “extremely tight?” The language that follows provides only a qualitative description of where the PIP₂ is located and its protein contacts. Describing it as “extremely tight” suggests a numerical precision that is not justified here. The language needs to be defended or, more likely, changed if such evidence cannot be stated.

Response: We have changed our description as suggested in line 177-178. PIP₂ binding is previously described as “extremely tight” based on its presence after membrane extraction and purification process, and the clear feature of the density, which indicates low flexibility of PIP₂ molecules in the apo state.

Line 199: It is stated here that, “Previous studies reported that transport activity of BTR1 is enhanced by intracellular alkalization.” However, in line 205 it is stated that, “Electrophysiology experiment results showed that the current densities of wild-type BTR1 hardly changed when pHi dropped from 7.4 to 7.0.” Although I agree that the subsequent data showing a current density for the R125H is significant, I see the two lines I quoted as a contradiction that is not acknowledged. What do the authors think is happening there?

Response: These studies reported that BTR1 could be activated by both intracellular and extracellular alkaline pH (Myers et al., 2016; Quade et al., 2020). The activity of BTR1 is obviously increased from pHi 7.03 to 7.40 at extracellular pH 8.5, but hardly changed from pHi 7.12 to 7.29 at extracellular pH 7.4 (Quade et al., 2020). In our electrophysiological experiments, cells were incubated at pHe 7.4 until the currents were stable under the whole-cell mode. Bath solution was then changed to pHe 8.0 for recording and the currents were stabilized in seconds. NH₄Cl stimulated currents could be reversed by the removal of NH₄Cl.

In their experiments, cells expressing BTR1 were incubated at pHe 7.4 or pHe 8.5 for several minutes and pHi were adjusted by clamping voltage to +20 mV (Myers et al., 2016) or by injection of 1.2 M NaHCO₃ solution (Quade et al., 2020). After adjustment of pHi and incubation of cells in buffer at pHe 8.5, NH₄Cl stimulated currents could not be reversed by removal of NH₄Cl (Myers et al., 2016). As we do not have evidence about the activating mechanism of pHe to BTR1, we can only hypothesize that short time exposure of cells to alkaline extracellular pH leads to the differences and our results may resembled their results at pHe 7.4.

Also, while the R125H data is nice, histidine can still retain a positive charge in near-neutral pH. Wouldn't it be reasonable to hypothesize that there would be an even more drastic (and pH-independent) difference when testing an R125A mutant? Did the authors ever construct such a mutant, and if so what did it show? If the authors never made such a mutant, then why not?

Response: R125A mutant was previously tested and its activity was not significantly reduced compared with that of R125H mutant. Therefore, we only show the effect of pathogenic mutation R125H. To answer the questions, we further tested the effect of R125E mutation. The significantly reduced activity of R125E mutant at pHe 8.0 supports our hypothesis.

Line 247: It should be noted here with appropriate citations that other SLC4 transporters have been proposed to act by the elevator mechanism. There is also an opportunity you should take here to compare the 4.23 angstrom shift you observe with those seen for elevator mechanisms. While yours is on the smaller magnitude

(compared to that of elevator transporters like GltPh), there are other proposed elevator mechanisms that have similar shifts and noting that context would be useful here.

Response: We have cited the references that propose SLC4 transporters to act by the elevator mechanism and listed two transporters that have similar displacement of the substrate binding site compared with BTR1 in line 254-264.

Line 317: More care must be taken to choose correct and proper citations. Arakawa et al (ref 17) did not propose SLC4A1 to use an elevator transport mechanism (the word “elevator” does not appear in that paper nor is such a mechanism described.) The first paper to report an elevator mechanism for SLC4 transporters and their homologs was Thurtle-Schmidt and Stroud (PMID: 27601653) and subsequently supported by Forrest and colleagues (PMID: 29167180) and Kurtz and colleagues (PMID: 36517642). The Arakawa reference should be removed from this line and the three references above should be inserted in their place.

Response: We have corrected the citation and inserted the references in line 337-340 as suggested.

Minor comments:

The title does not read grammatically correct and needs to be changed. In particular, “with PIP2 and conformational change” has nonparallel structure.

Response: We have changed the title to “Structural insights into the conformational change of BTR1/SLC4A11 in complex with PIP₂” as suggested.

There are numerous word choices that need to be fixed for grammar or readability. For example line 27 should be “activation” not “activated.” In lines 148-149 “which is likely account for” is incorrect word choice. There are many instances like these in the manuscript which will require close inspection to correct. Additionally, check for spelling throughout the entire paper. Words like “pipette” are misspelled.

Response: We have made these corrections in the revised manuscript.

Line 62: The phrase, “which severely reduces the quality of life of affected individuals” is not needed and should be deleted. Some might find that language ableist.

Response: We have deleted the phrase as suggested.

Line 74: structures do not “fail;” rather simply state that they “did not resolve...”

Response: We have made the correction as suggested.

Line 378: state that the purpose of the TEV protease was, presumably, to remove the GFP and His-tag from the protein (this is never stated anywhere.)

Response: We have added the purpose of TEV protease incubation in line 399-400.

Line 423: Be more precise than stating “some residues are still missing.” Which residues are absent from the model?

The formatting of citation 12 has degree information included and needs to be fixed.

Response: We have made the correction of citation 12 and included detailed information of the model content in the revised Supplementary Table 1.

Reviewer #2 (Remarks to the Author):

This manuscript reports cryo-EM structures of human SLC4A11 transporter in several different conditions. The structures appeared to be of high quality, with resolutions ~3.0 angstrom. The structures revealed several interesting aspects about this family of transporters, including identification of a presumed PIP₂ binding site, drastic conformational changes in the cytoplasmic domain, and potentially distinct conformations along the transport cycle. However, the structural and functional analysis and interpretation are not well presented. Overall, the manuscript would need significant improvement in terms of scientific accuracy as well as editing to improve grammar and readability. Below are some major points.

1. Page 6-7, substrate binding sites. The authors claimed densities corresponding to NH₃ molecules in Supplementary Fig. 3h and 3i. A comparison of the cryo-EM densities in this region in the apo and NH₄Cl structures would be important.

Response: Densities along the ion permeation pathway are only putative water and ammonia molecules as they are difficult to be distinguished. A comparison of the cryo-EM densities have been shown in Supplementary Fig. 3i and 3j for better visualization of the differences between these two states as suggested.

2. PIP₂ binding and functional consequence. Several basic residues, such as R128 and R227, have been identified to coordinate the PIP₂ headgroup but mutations R128A and R227A showed activity similar to the wild type (Fig. 3f). What is the explanation?

Response: The negligible effects of R128A and R227A mutations on BTR1 activity may result from their weak interaction with phosphate groups of PIP₂ molecule and residues at the intracellular loop of TMD. Positive charged side chains of R125 and K263 interact with both two phosphate groups of PIP₂ in the model of BTR1_{OF/APO}. K260 interacts with the inner inositol head and forms hydrogen bond with Q812 in

the intracellular loop of TMD. R227 only interacts with the outer inositol head of PIP2 and R128 locates relatively far from both inositol head groups (~5 Å and ~8 Å).

3. On page 11, structural comparison of BTR1 (outward-facing/APO) and BTR1 (inward-facing/R125H). The authors aligned these two structures by the 'gate' domain and then claimed that the 'gate' domain remains relatively unchanged (RMSD 1.678 Å) and the 'core' domain undergoes apparent changes (RMSD 3.784 Å). This is problematic. What happens if the 'core' domain is used to align these two structures? Are the 'gate' domain and the 'core' domain undergoing rigid-body motions? More structural analysis, such as comparison of domain movements within a protomer and between dimers, would help understand structural rearrangements in these two conformations.

Response: We compared the two states by aligning the gate domains based on their rigidity during conformation change. Structural comparison of TMD by aligning the core domain, RMSD of the core domain and the gate domain are 3.318 Å and 3.094 Å, respectively. The gate domain (RMSD 0.547 Å) and the core domain (RMSD 1.093 Å) are rigid during conformation transition. We have aligned the two states and described in line 254-257 and Supplementary Fig. 6b and 6c as suggested. Comparison of domain movements within a protomer is shown in Fig. 4g and 4h. Movements between dimers are shown in Supplementary Fig. 6e as suggested.

4. Assignment of outward and inward facing conformations. From Figs. 2d and 4d, the openings at the cytoplasmic side for the assigned outward- and inward-facing conformations are similar. How do the authors assign the functional states of these two conformations to outward- and inward- facing, respectively?

Response: Inward-facing and outward-facing conformation are mainly assigned by aligning with other SLC4 family transporters in the outward-facing state (Supplementary Fig. 3a-c) and the movement of core domain between two states of BTR1.

5. Elevator mechanism. From structural comparisons and Fig. 6, it appears that the 'core' domain rocks against the 'gate' domain to generate alternate access. This would not fit with an elevator type of transport.

Response: The core domain and gate domain are rigid during the conformation transition process of BTR1 (Supplementary Fig. 6b and 6c). The substrate binding sites are mainly formed by the core domain (Fig. 2e). The core domains move vertically relative to the membrane during state transition from outward-facing to inward-facing (Fig. 4h). These features fit the elevator-like transport mechanism better.

6. It seems that the complete transport pathway is within each subunit of the dimeric transporter. Do mutations at the dimer interface (the 'pore' domain and NTD) alter transport function?

Response: There are several mutations at the dimer interface have been reported to be pathogenic. For example, Nonsense mutations R112X and R605X both leads to truncation of BTR1 and CHED2 (Sultana et al., 2007). In addition, V575M and G583D are related with Fuchs corneal dystrophy (FCD) (Riazuddin et al., 2010). G583D mutation leads to trapping in the ER and degradation of mutant protein compared with wild-type BTR1. In contrast, V575M exhibits partial loss of localization at the membrane and is likely to alter transport function. Moreover, C611R has been identified to impair the protein function and causes CHED2 (Kodaganur SG, 2013).

Reference:

Kodaganur SG, K.S., Veerappa AM, Tontanahal SJ, Sarda A, Yathish S, Prakash DR, Kumar A (2013). Mutation analysis of the SLC4A11 gene in Indian families with congenital hereditary endothelial dystrophy 2 and a review of the literature. *Mol Vis* 19, 1694-1706.

Myers, E.J., Marshall, A., Jennings, M.L., and Parker, M.D. (2016). Mouse Slc4a11 expressed in *Xenopus* oocytes is an ideally selective H⁺/OH⁻ conductance pathway that is stimulated by rises in intracellular and extracellular pH. *Am J Physiol Cell Physiol* 311, C945-C959.

Quade, B.N., Marshall, A., and Parker, M.D. (2020). pH dependence of the Slc4a11-mediated H⁽⁺⁾ conductance is influenced by intracellular lysine residues and modified by disease-linked mutations. *Am J Physiol Cell Physiol* 319, C359-C370.

Riazuddin, S.A., Vithana, E.N., Seet, L.F., Liu, Y., Al-Saif, A., Koh, L.W., Heng, Y.M., Aung, T., Meadows, D.N., Eghrari, A.O., *et al.* (2010). Missense mutations in the sodium borate cotransporter SLC4A11 cause late-onset Fuchs corneal dystrophy. *Hum Mutat* 31, 1261-1268.

Sultana, A., Garg, P., Ramamurthy, B., Vemuganti, G.K., and Kannabiran, C. (2007). Mutational spectrum of the SLC4A11 gene in autosomal recessive congenital hereditary endothelial dystrophy. *Mol Vis* 13, 1327-1332.

Reviewers' Comments:

Reviewer #1:

Remarks to the Author:

Having had the chance to review the revised manuscript, "Structural insights into the conformational change of BTR1/SLC4A11 in complex with PIP₂," the paper has improved from the last version as many of the comments have been addressed. But there is one important issue I think the authors did not properly resolve:

Mainly, reporting a substrate-bound structural state is of high importance to a field, but can be misleading or even disastrous when the substrate is misplaced or misidentified. In my prior evaluation, I asked the authors to clarify by what basis they were placing ammonia molecules into their experimental density. The other reviewer asked a similar question. In their reply, the authors equivocate that the densities along the pathway are "only putative H₂O and NH₃ molecules," seemingly acknowledging that they cannot be certain of which molecule is which given the densities. However, in the actual text in lines 136-138 the authors write: "we identified two densities with relatively different intensity from the BTR1OF/APO state in the substrate binding sites, which are likely NH₃ molecules (Supplementary Fig. 3h-3j)." This language has now shifted from "putative H₂O and NH₃" to "likely NH₃." Examining the Supplemental Fig. 3h-3j makes things worse, because in panels "h" and "i" they show densities as blue or red and indicate that they represent ammonia and water, respectively. There is no basis offered in either the caption or in the methods section for why they assign these peaks with different identities. The closest the authors come to offering a justification is when they write in line 152: "Am1(ammonia molecule 1) and Am2 both locate at the substrate binding pocket and Am1 forms hydrogen bond with the backbone carbonyl group of T434." The reasons boil down to (a) they are in a reasonable location, and (b) one of them can hydrogen bond with T434. But this is not persuasive because (a) the correct location can be in an empty apo state, which happens frequently, and (b) water can also make a hydrogen bond with a carbonyl group in T434, so it is not clear why that should be convincing. Although it is plausible that one or more of these densities could be ammonias, I do not think the data justifies making that conclusion with the level of certainty currently stated in the revised text. Thus I think the authors need to be clear, in all instances including the text, figures, captions, and supplement, that while there are difference densities in the solute translocation pathway, and they could be either waters or ammonia, the resolution and detail of the densities precludes making those identifications with certainty.

I do not think this manuscript hinges on the ammonia assignment, and that it would take away little to make the claim as I described.

Reviewer #2:

Remarks to the Author:

The manuscript is improved, and my major concerns are addressed.

Reviewer #1 (Remarks to the Author):

Having had the chance to review the revised manuscript, “Structural insights into the conformational change of BTR1/SLC4A11 in complex with PIP2,” the paper has improved from the last version as many of the comments have been addressed. But there is one important issue I think the authors did not properly resolve:

Mainly, reporting a substrate-bound structural state is of high importance to a field, but can be misleading or even disastrous when the substrate is misplaced or misidentified. In my prior evaluation, I asked the authors to clarify by what basis they were placing ammonia molecules into their experimental density. The other reviewer asked a similar question. In their reply, the authors equivocate that the densities along the pathway are “only putative H₂O and NH₃ molecules,” seemingly acknowledging that they cannot be certain of which molecule is which given the densities. However, in the actual text in lines 136-138 the authors write: “we identified two densities with relatively different intensity from the BTR1OF/APO state in the substrate binding sites, which are likely NH₃ molecules (Supplementary Fig. 3h-3j).” This language has now shifted from “putative H₂O and NH₃” to “likely NH₃.” Examining the Supplementary Fig. 3h-3j makes things worse, because in panels “h” and “i” they show densities as blue or red and indicate that they represent ammonia and water, respectively. There is no basis offered in either the caption or in the methods section for why they assign these peaks with different identities. The closest the authors come to offering a justification is when they write in line 152: “Am1 (ammonia molecule 1) and Am2 both locate at the substrate binding pocket and Am1 forms hydrogen bond with the backbone carbonyl group of T434.” The reasons boil down to (a) they are in a reasonable location, and (b) one of them can hydrogen bond with T434. But this is not persuasive because (a) the correct location can be in an empty apo state, which happens frequently, and (b) water can also make a hydrogen bond with a carbonyl group in T434, so it is not clear why that should be convincing. Although it is plausible that one or more of these densities could be ammonias, I do not think the data justifies making that conclusion with the level of certainty currently stated in the revised text. Thus I think the authors need to be clear, in all instances including the text, figures, captions, and supplement, that while there are difference densities in the solute translocation pathway, and they could be either waters or ammonia, the resolution and detail of the densities precludes making those identifications with certainty.

I do not think this manuscript hinges on the ammonia assignment, and that it would take away little to make the claim as I described.

Thanks for the review and we have revised the manuscript and Supplementary Figure. 3 as suggested. Densities along the substrate permeation pathway are indicated by Molecule 1-6 (M1-6) without defining the identities. Two densities with relatively different intensity between the two states are indicated by M5 and M6 in Supplementary Fig. 3h-j.

Reviewer #2 (Remarks to the Author):

The manuscript is improved, and my major concerns are addressed.

Thanks for the review and the positive feedback.